# Hidden network preserved in Slide-tags data allows reference-free spatial reconstruction

**Simon K. Dahlberg** [ID], **David Fernández Bonet** [ID], **Lovisa Franzén** [ID], **Patrik L. Ståhl** [ID] [✉] **& Ian T. Hoffecker** [ID] [✉]

Spatial transcriptomics technologies aim to spatially map gene expression in tissues and typically use oligonucleotide array surfaces that have undergone spatial indexing. These arrays are used to capture nucleic acids diffusing from adjacently placed tissues, allowing subsequent sequencing to reveal both gene and position. Slide-tags is a recently developed method by Russell et al. that inverts this principle. Instead of capturing molecules released from the tissue, probes are detached from a pre-decoded bead array and diffused into tissues, tagging nuclei with spatial barcodes. In this work we reanalyze this data and discover a latent, spatially informative cell-bead network formed incidentally from barcode diffusion and the biophysical properties of the tissue. This allows us to treat Slide-tags as a network-based imaging-by-sequencing approach. By optimizing spatial constraints encoded in the cell-bead network structure, we can achieve unassisted tissue reconstruction, a fundamental shift from classical spatial technologies based on pre-indexed arrays.

Sequencing-based spatial transcriptomics technologies[1–6] aim to localize gene expression in tissue while preserving broad molecular coverage. These methods depend on spatially patterned arrays of barcoded oligonucleotides to capture transcripts from adjacent tissue sections. Spatial positions must be recovered through a decoding step, either via deterministic assignment during array fabrication[2,4,6] or by in situ fluorescence-based sequencing of randomly dispersed arrays[3,7,8].

Russell and colleagues recently developed Slide-tags[1], a conceptual inversion of the classical array capture principle. Instead, spatially decoded bead arrays are cleaved to diffuse oligonucleotide barcodes into tissue samples, thus associating spatial barcodes with individual nuclei. This approach maintains compatibility with single-nucleus RNA sequencing and cleverly exploits diffusion to associate known spatial barcodes with biological targets. However, like other spatial array-based approaches, this method still requires a spatial reference map obtained prior to the experiment, upholding the central dependency on a prior decoding step shared across spatial transcriptomics methods (Fig. 1a upper path).

Although Slide-tags aims to tag the nearest cell to each bead, we recognized that diffusion is inherently imprecise. Barcodes from a single bead can reach multiple nearby cells, and conversely, multiple neighboring beads are likely to be connected to multiple nearby cells. This many-to-many pattern of associations corresponds to a bipartite network, where barcoded beads and nuclei represent two distinct node classes, bead-nodes and cell-nodes, connected by edges representing tagging events. We realized that the structure of this network reflects the physical layout of the tissue. Since cell-bead connections are more likely to form when elements are spatially close, the network satisfies the criterion of a spatial network in which distance relationships determine connectivity. In this work, we developed an alternative pipeline (Fig. 1a lower path) that interprets Slide-tags data as a spatial cell-bead network and infers cell positions directly from its structure, even in the absence of explicit coordinates provided by a prior reference map.

We previously developed an algorithm, Spatio-Topological Reconstruction by Network Discovery[9] (STRND; Fig. 1b), to extract positions from spatial networks. STRND models each node's position based on its random walk visitation profile, or how frequently other nodes are encountered from a given starting point. These profiles are captured as high-dimensional data that, when projected into two dimensions using dimensionality reduction techniques, yields an

Department of Gene Technology, KTH Royal Institute of Technology, Science for Life Laboratory, Solna, Sweden. [✉]e-mail: patrik.stahl@scilifelab.se; ian.hoffecker@scilifelab.se

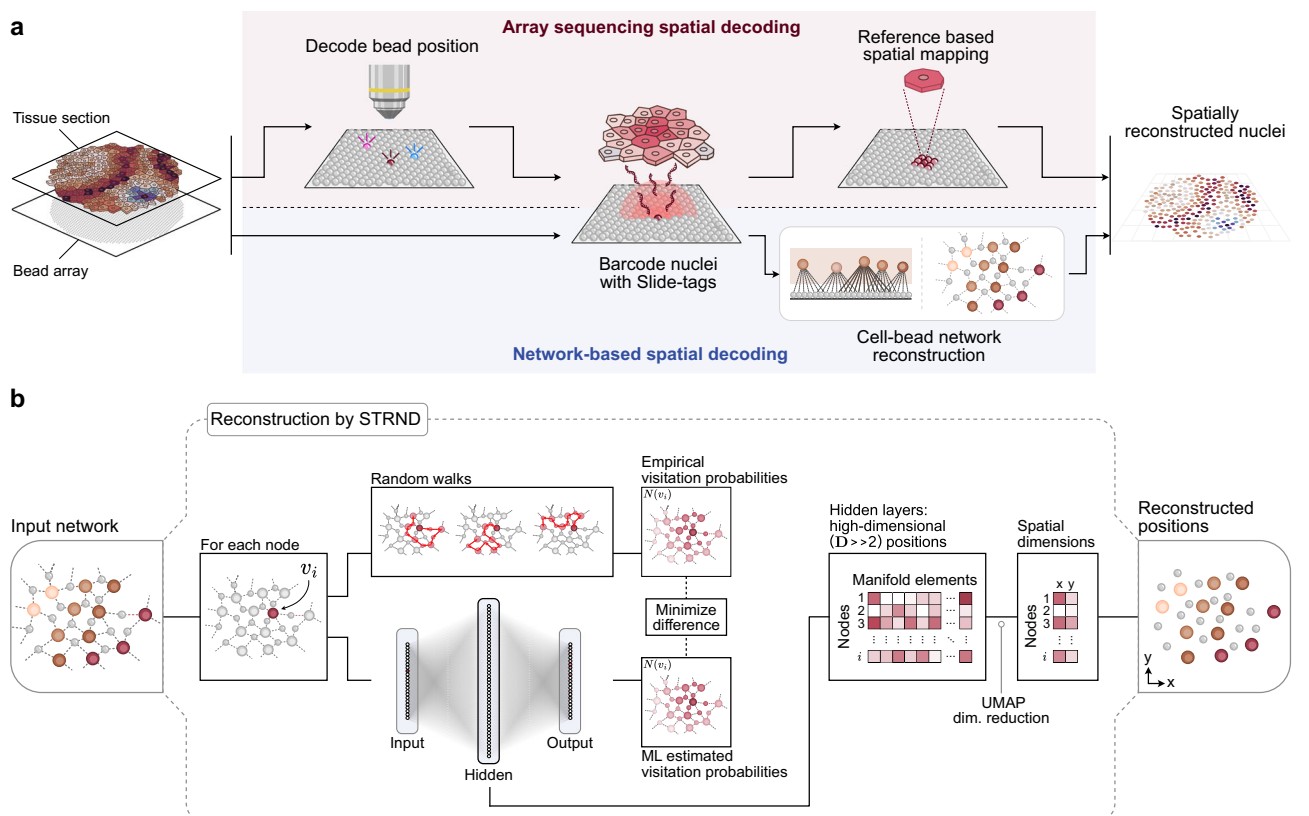

**Fig. 1 | An alternative, cell-bead network pipeline for spatial reconstruction from Slide-tags data. a** Schematic of the original Slide-tags workflow (top path) in which optical decoding is used to obtain a spatial reference map, and a proposed alternative workflow (bottom path) that exploits an incidental cell-bead network formed during the experiment to extract latent spatial information in the network structure. **b** Schematic of the STRND reconstruction algorithm used to recover spatial cell positions. The algorithm begins with an abstract network as input and conducts a series of random walks from each node. Each node's position in the network is uniquely defined by its random walk visitation probability neighborhood, which for spatial networks, may be reduced to 2 spatial dimensions as the output of the pipeline.

approximation of the nodes' spatial positions. Thus, diffuse barcode spread, rather than a source of imprecision, becomes an informative pattern that encodes local geometry. This reinterpretation of Slide-tags would place it within the emerging field of imaging-by-sequencing, also known as DNA microscopy, molecular pixelation, or sequencing-based microscopy[10–18] in which, even in the absence of a prior spatial reference map, spatial positions of tissue elements are located based on sequencing data alone.

## Results

### Latent spatial relationships in the cell-bead association network
We examined three data sets from Russell et al.'s work: a human tonsil sample and two mouse samples; embryonic brain and adult hippocampus (Fig. 2a). We first examined basic network properties starting with the distribution of the number of cell-bead edges per cell, i.e., the degree distribution. In all three cases, the degree distribution exhibited a bounded, unimodal distribution with a definable mean and spread. This aligns with what we might expect from a spatial network, in which each node forms connections with a limited number of nearby neighbors due to physical proximity constraints, rather than arbitrarily distant or numerous nodes as occurs in scale-free networks. Multiple cells in a spatial network should also share edges to the same beads, as beads should connect to multiple cells in their proximity. Accordingly, the mean degree for each sample is significantly higher than what would be the case if each bead only connected to a single cell (Fig. 2b, Supplementary Fig. 1) indicating the type of overlap that would be a prerequisite for a continuous spatial network.

To further investigate whether the cell-bead network reflects spatial organization, we projected the (bipartite) cell-bead network onto only its cell-nodes, converting it into a (unipartite) cell-cell network. In this projection, two cell-nodes are connected if they share at least one bead-node neighbor and all bead-nodes and their edges are then removed from the network. Doing this also allowed us to inspect indirect connections between cell-nodes along with their respective Russell et al.-determined reference positions to assign a physical distance between cell-nodes in the projected network. We examined the distribution of cell-cell edge distances (normalized by the expected edge length frequencies in a fully connected network). In all three samples, we observed a population of edges in the short, 0–200 μm range whose frequencies were elevated compared to the longer edges (Fig. 2c, Supplementary Fig. 2). This indicates that cells that are close to each other in physical space are more likely to share a mutual bead than cells that are further apart. This would, by extension, suggest that there is a significant subpopulation of cell-bead edges formed by a proximity-dependent diffusion mechanism and that those beads connect to multiple cells within their respective proximities.

To visualize the spatial characteristics of the cell-cell network, we overlaid the cell-cell edges derived from downstream filtering steps (see next sections and Methods) onto the cell-node reference positions reported by Russell et al. Here we observed an apparent abundance of short-range (i.e., spatially formed) connections (Fig. 2d, e). These indicators of a continuous, spatially formed, and sample-spanning network suggest that it should be possible to recreate the physical spatial relationships that led to its formation. If the proportion of spatially formed edges in relation to the noise-derived or non-

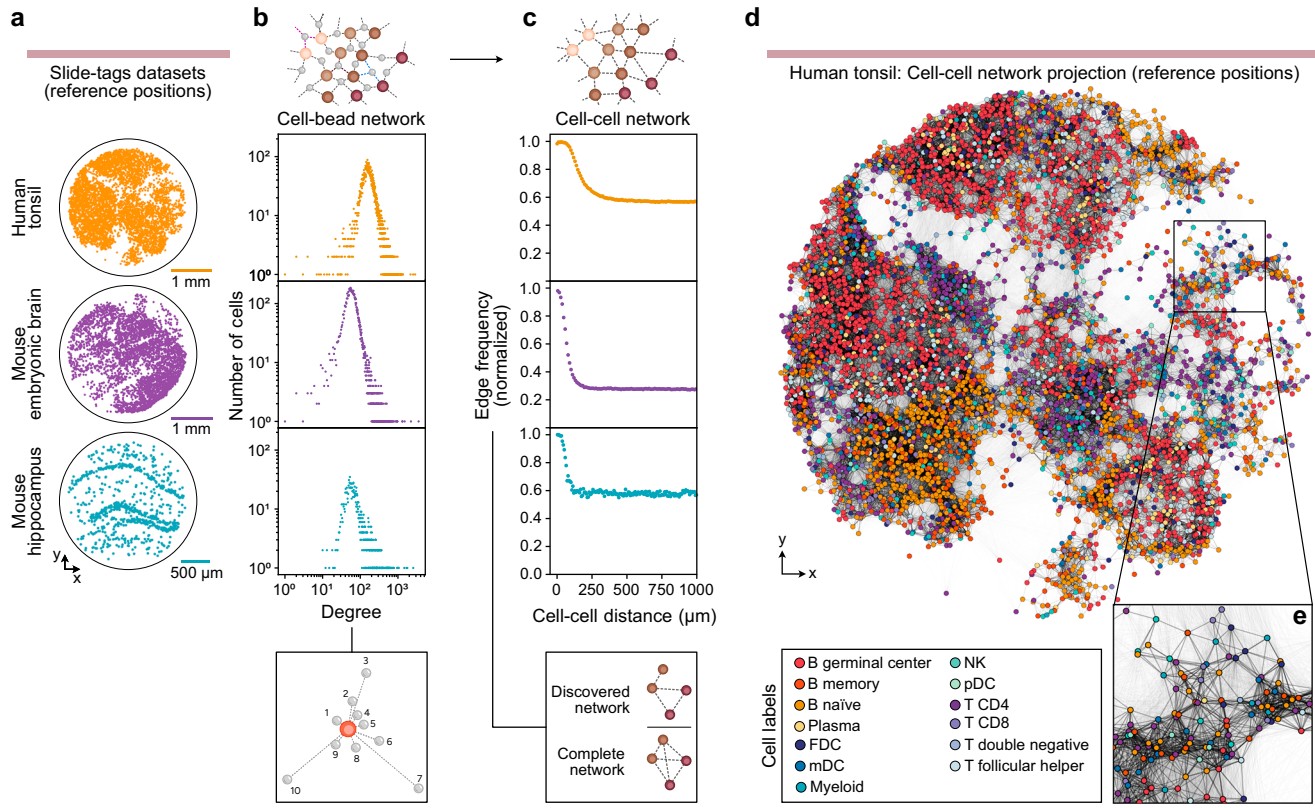

**Fig. 2 | Slide-tags data contains a hidden network of spatially linked cells.**
**a** Publicly available reference positions of cells in the biological samples with bead-cell networks available, with the relevant bead-cell sequencing data for the tonsil sample being directly supplied by Russell et al. and each respective mouse having fully public sequencing data including bead-cell sequencing data. **b** Degree distributions for the cells of the three biological samples respectively, all showing distinct, unimodal distributions as expected of a network where a node's number of connections are constrained by physical distance. **c** The frequency of edges in the

cell-cell network (unipartite projection) as a function of cell-cell distance, normalized by the total number of possible edges for each distance interval.
**d** Visualization of cell-cell network (unipartite projection): reference positions of the tonsil sample colored by cell type with cell-cell edges from the fully processed network (see methods). **e** Closeup inset of the network in e, highlighting the presence of a subpopulation of short-range edges consistent with spatial formation. Source data are provided as a Source Data file.

spatially correlated edges is sufficient, an algorithm like STRND could use network structure alone to place cell and bead nodes in space so as to satisfy their proximity constraints defined by their edges.

**Initial reconstruction reveals sample-dependent spatial signal**
The three samples exhibited significant differences in their respective network properties. Within each sample network, the tonsil and embryonic brain samples have a similar number of cells at 9352 and 9015, respectively, while the hippocampus sample had markedly fewer cells with only 1658. For each sample, approximately half of the cells were present in the original Slide-tags reference maps, at 5777 (62%), 4622 (51%), and 839 (51%) for the tonsil, embryonic brain, and hippocampus samples, respectively. In terms of total network connectivity, however, the tonsil sample exhibited greater density than either mouse sample at 1,949,184 cell-bead edges, or more than twice that of the embryonic brain sample with 708,611, and more than ten times that of the hippocampal sample with only 188,086 (Fig. 3a, Supplementary Fig. 3).

Given the substantial differences in network properties across the three samples, we next performed an initial single-pass network-based (STRND) reconstruction of the full cell-bead (bipartite) networks without filtering, aiming to reconstruct spatial organization without any handling of noise or preprocessing. Because network-based reconstruction does not preserve scale, rotation, and mirroring, we assessed performance using two complementary metrics based on relative spatial relationships: one based on global structure and one for local neighborhood preservation. A global fidelity metric quantifies

large-scale geometric consistency by computing the Pearson correlation coefficient between pairwise distances in the reconstructed and reference positions (Correlated Pairwise Distances, CPD; Methods, Supplementary Fig. 4a). The local fidelity metric measures the proportion of shared nearest neighbors for each node by comparing its $K$ = 15 nearest neighbors in a reconstructed space to those in the reference layout (KNN score; Methods, Supplementary Fig. 4b). To account for the stochastic nature of STRND, we repeated the single-pass reconstruction 10 times per sample and report the mean and standard deviation of each metric. Since both metrics rely on ground-truth reference information, only cells with spatial coordinates in the original Slide-tags analysis (5777 for tonsil, 4622 for embryonic brain, 839 for hippocampus) contributed to the evaluation.

Single-pass reconstructions of the tonsil sample significantly outperformed both mouse samples in both metrics. For global structure, the tonsil network yielded a mean CPD $R^2$ of 0.758±0.041, while the embryonic brain and hippocampus showed far lower values (0.082±0.002 and 0.040 ± 0.011, respectively; Fig. 3b left, Supplementary Fig. 5a-c). This suggests that the tonsil reconstruction preserves large-scale spatial features such as relative cardinal orientation and tissue-scale organization, whereas the mouse samples lack globally coherent structure. Local fidelity of single-pass reconstruction followed a similar pattern, though with less marked differences. The tonsil sample again outperformed the others, with a mean KNN score of 0.350 ± 0.006 compared to 0.084 ± 0.028 and 0.146 ± 0.007 for mouse embryonic brain and hippocampus, respectively (Fig. 3b right, Supplementary Fig. 5d–f). These results indicate that the tonsil

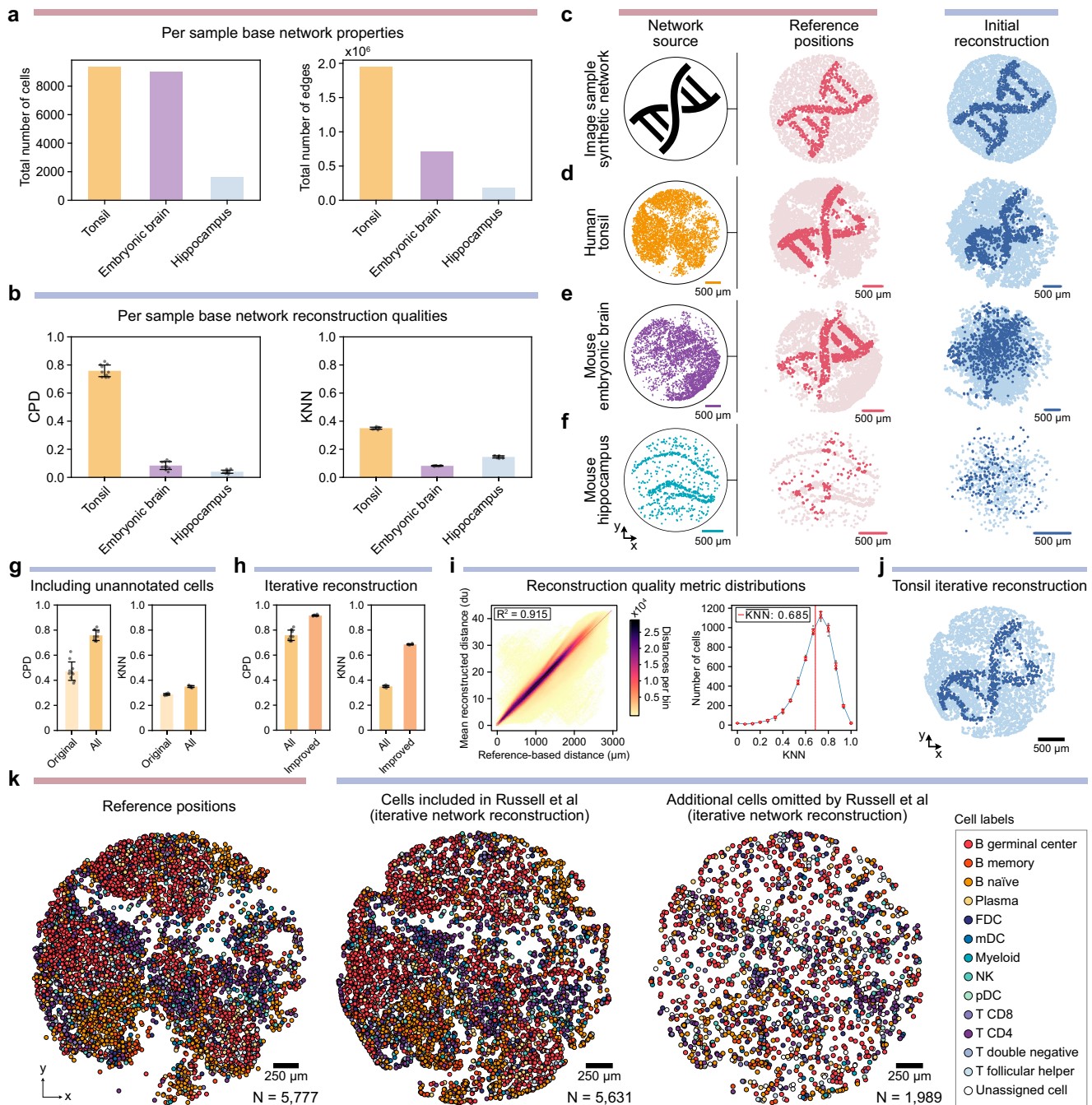

**Fig. 3 | Optimization of proximity constraints enables cell localization without prior spatial indexing. a** Sample comparison with respect to number of in-network cells (left) and total number of cell-bead edges (right). **b** Metrics quantifying the reconstruction quality by comparing to the reference positions using the correlated pairwise distances (CPD) (left), and mean overlap between K nearest neighbors (KNN) (right) after 10 single-pass STRND reconstructions of each network. **c** Artificial data reference positions (left) and network-based reconstruction (right). First reconstruction (n = 1 out of N = 10) from **b** chosen as representative reconstruction for each sample with each node's color shade sampled from the same image as the synthetic example, showing (**d**) the tonsil sample is largely recreated albeit with inaccuracies, (**e**) the mouse embryonic brain sample has lost most local structure, and (**f**) the mouse hippocampus sample shows little discernible remaining structure. **g** The improvement in the two reconstruction quality metrics when using not only cells with reference positions (Original) but also remaining cells present in the network (All). **h** Improvement in global and local reconstruction quality metrics after iterative reconstruction and morphing

(Improved). **i** Correlated pairwise between reference position distances and mean reconstructed distance over 10 reconstructions (N = 5631 Cells, N = 15,851,256 distances), colored by number of points in each of 10,000 bins (left) and the distribution of overlapping K-nearest neighbors over N = 10 reconstructions, with the vertical dashed line showing the total mean value and the full line showing the per value mean ± SD (right). **j** The reconstruction of the cell-bead network generated from the tonsil sample after iterative reconstruction improvements, showing the improved structure compared to **d**. **k** Visualization of reference positions (left), network-based reconstructed positions for originally included cells (middle), and additional cells (right) colored by label-transfer cell type cluster. Data in bar graphs are presented as mean ± SD over N = 10 reconstructions of the network relevant to each bar, characterizing technical variance in STRND reconstruction. Reconstruction improvements in **g** and **h** show complete separation (Wilcoxon statistic = 0.0, p-value = 1.83e-4) indicating the improvement is not derived from STRND stochasticity. Source data are provided as a Source Data file.

network retains both global and local spatial information to a far greater extent than the mouse networks. Together, the metrics show differences in initial reconstruction quality that likely reflect underlying differences in the spatial integrity of the networks, such as differences in numbers of edges generated by a proximity-dependent spatial process versus noise or edges not reflecting any spatial correlation.

To complement quantitative reconstruction metrics, we applied a visual approach to assess reconstruction quality at both global and local scales in which we projected the colors of a distinct reference image (a stylized double helix icon) onto the spatial positions of cells with known reference coordinates, and then observed how these colors were preserved following the single-pass network reconstruction. For a baseline demonstration of an ideal spatial network, we first applied this approach to a synthetic spatial network in which points were randomly placed over the image and assigned a color based on positions, and connected to their neighbors using a contact-based tessellation. Reconstructing such a synthetic network with STRND yields an accurate recovery of the original image, with clear preservation of relative positions and spatial structure (Fig. 3c).

In the biological datasets, we applied the same image projection approach to cells that have associated reference spatial positions which we used to approximate ground truth. For the tonsil sample, the reconstructed image retains key features of the original pattern, albeit with some distortion and local noise. This visual coherence corroborates the high quantitative reconstruction scores and indicates that the tonsil network contains a strong spatial signal. (Fig. 3d). In contrast, the mouse embryonic brain reconstruction preserved only vague traces of the original image, with some clusters of similarly colored nodes located roughly in the vicinity of one another, and overall structure scrambled or consistent with the lower global and local reconstruction metrics (Fig. 3e). The hippocampus sample reconstruction exhibited even greater loss of its original structure (Fig. 3f).

The significant differences in spatial network quality across samples, evident both in connectivity statistics and reconstruction metrics, motivate different reconstruction strategies. In the tonsil network, the density and quality of the cell-bead associations are high enough that, in a single-pass reconstruction of the entire sample, spatial layouts retained global structure (Fig. 3d). The richness of this network makes it amenable to iterative filtering: noisy or weakly supported edges can be pruned without compromising overall integrity. In contrast, the mouse embryonic brain and hippocampus samples exhibit sparser and noisier networks, where full-sample reconstruction failed to recover spatial structure on a single-pass. In these cases, identification of localized regions of high connectivity and spatial coherence could be extracted as fragmentary subgraphs and reconstructed individually (Supplementary Fig. 6), allowing partial recovery.

## Iterative refinement and cell inclusion improves reconstruction performance

Since reconstruction quality metrics depend on comparing inferred and reference positions, only cells with reference coordinates can be directly assessed. As a result, we initially limited reconstruction to this subset of cells, assuming they represent the highest-quality spatial data. However, we found that including all in-network cells, regardless of whether they had associated reference positions, substantially improved both global and local reconstruction metrics (Fig. 3g). This suggests that even cells excluded from the original Slide-tags analysis still contribute spatially relevant information, with the improvement reflecting a cumulative effect in which additional cells provide redundant proximity information that reinforces consistent spatial structure in the network.

To further improve reconstruction quality, we applied a post hoc filtering step inspired by the original Slide-tags pipeline. Using the reconstructed positions, we applied DBSCAN clustering to identify and remove long-range edges representing noise or unlikely to be reflective of spatial correlation. This reduced the influence of non-local forces that tend to compress and distort spatial positions, resulting in a clearer second-pass reconstruction (Methods, Supplementary Fig. 7). We also incorporated a soft morphing step exploiting the available (i.e., not privileged) geometric knowledge of the tonsil sample's disc shape (Methods, Supplementary Fig. 8) in which gentle adjustment of the layout after reconstruction resulted in a slight improvement in global reconstruction accuracy.

These refinements together result in substantial improvements in reconstruction quality. The global CPD metric increased from 0.758±0.041 to 0.915±0.006, and the local KNN score nearly doubled from 0.350 ± 0.006 to 0.685 ± 0.003 (Fig. 3h-i, Supplementary Fig. 9). Notably, the large improvement in local accuracy suggests that long-range spurious edges disproportionately disrupt neighborhood structure. Visual inspection of the reconstruction with an image projection further confirmed the preservation of structural features, qualitatively surpassing the clarity exhibited in the single-pass reconstruction (Fig. 3j).

The ability to generate a high-quality initial reconstruction that could then be iteratively refined thus ultimately yielded a near-complete, reference-free reconstruction of the tonsil sample. This refined reconstruction recovered the spatial arrangement of 5631 out of 5777 (97%) cells with reference positions and additionally recovered 1989 cells that had been excluded from the original analysis. In total, this corresponds to 81% of all 9352 in-network cells being assigned spatial coordinates (Fig. 3k). This result demonstrates not only the robustness of the underlying spatial signal in the network, but also the capacity for the method to recover and spatially resolve cells that would otherwise be discarded, expanding the usable content of the dataset.

## Spatial reconstruction of human tonsil data preserves the biological context

To evaluate whether the network-based spatial reconstruction of the Slide-tags tonsil data preserves biologically meaningful spatial context, we applied a series of standard downstream analyses that depend on spatial information. A common application of spatial transcriptomics is to assess the colocalization and segregation of specific cell types, which can reveal functionally relevant tissue organization.

We began by visually inspecting the reconstructed spatial distributions of selected cell type pairs. The reconstructed positions successfully recapitulated key spatial features of the human tonsil, including separation of germinal center B cells and naïve B cells, as well as the colocalization of CD4$^+$ T cells and natural killer (NK) cells in the T cell zone (Fig. 4a). To systematically compare cell type spatial relationships between the reference and reconstructed coordinates, we performed neighborhood enrichment analysis, quantifying pairwise colocalization across all annotated cell types. The reconstructed positions of the cell types captured equivalent trends as observed in the reference data, with a Pearson correlation of 0.996, which would lead to similar biological conclusions from the data (Fig. 4b). To extend the analysis beyond the immediate neighboring cells, we computed co-occurrence probability scores that span a larger distance from the cell type of interest. Again, the reconstructed coordinates mirrored the reference data, capturing consistent spatial organization patterns (Fig. 4c).

We next applied Ripley's L function, a measure of spatial point patterning that quantifies clustering or dispersion across spatial scales. Computing the L-function for cells with both reference and reconstructed spatial positions yielded nearly identical results (Fig. 4d), with the area under the curve for all cell types showing a median difference of only ~2% (Fig. 4e). This further confirms that the network-based spatial reconstruction preserves both local and global spatial tissue structure to an extent that allows appropriate biological conclusions to be drawn from the data.

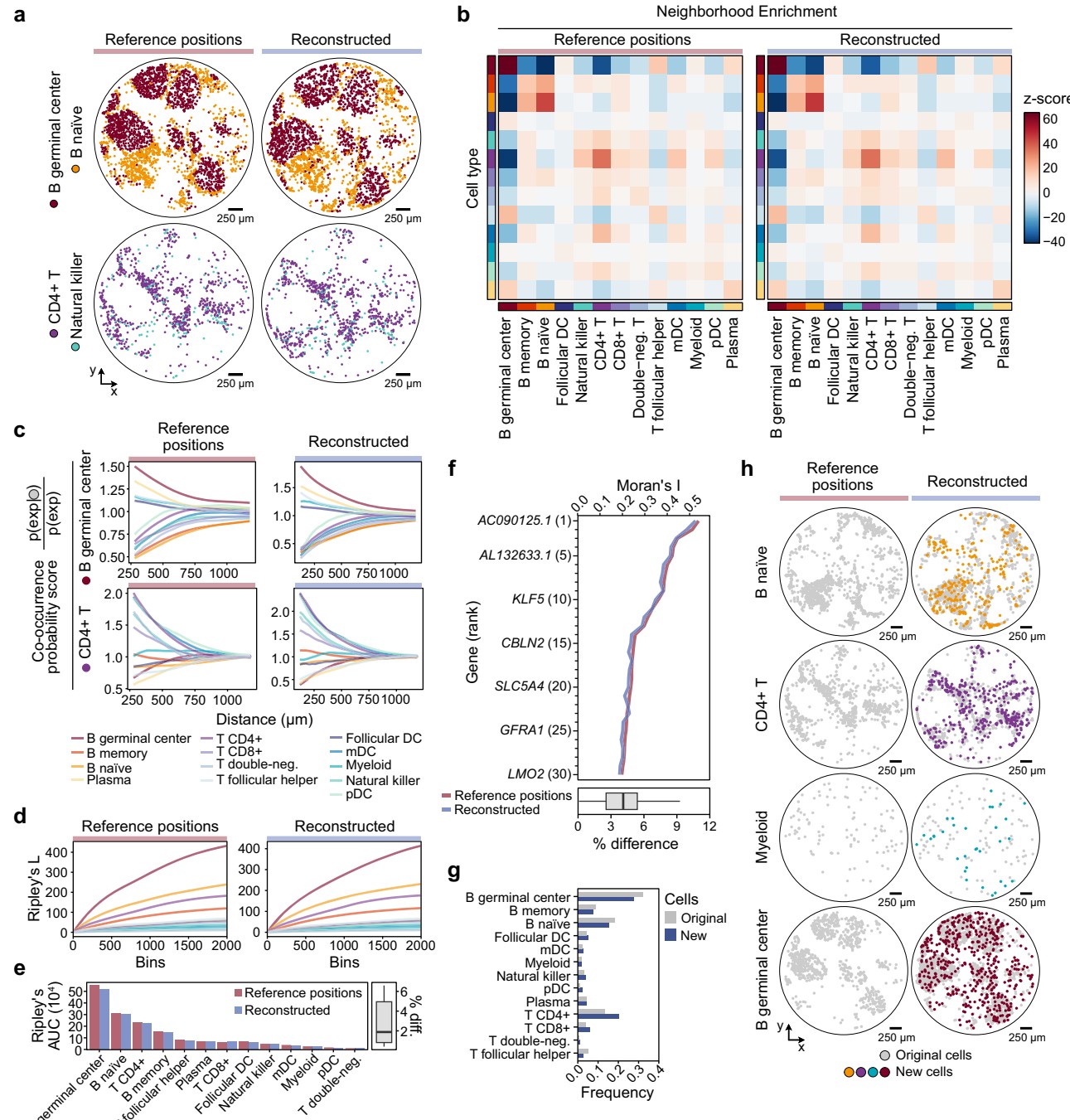

**Fig. 4 | The reconstructed human tonsil network accurately recovers the biological context. a** Comparison of reference positions and reconstructed spatial positions of cells in the human tonsil dataset, showing the pairwise co-localization of selected cell types. **b** Neighborhood enrichment z-scores for all cell types, performed for reference and reconstructed cell positions separately. High z-scores signify co-localizing cell types, while strong negative scores show spatial separation. **c** Co-occurrence probabilities in reference positions and reconstructed data for two selected cell types: B germinal center cells and CD4 + T cells. The co-occurrence score is computed for each cell type (exp) given the conditional probability of co-localizing with the cell type of interest (gray circle), and across increasing radii around the cells. **d** Ripley's L statistic for reference and reconstructed spatial positions, comparing the spatial distribution of each cell type, considering increasing distance bin sizes. Cell types with high scores correspond to aggregated spatial patterns, while low scores are assigned to more dispersed cell types. **e** By computing the area under the curve (AUC) for all cell types presented in

**d**, the scores can be more directly compared between the reference and reconstructed positions (right, barplot). Summarizing the difference (diff.) in scores as percentages for the $N = 13$ cell types (left, boxplot) shows a median difference of approximately 4%. **f** Top 30 most spatially variable genes, based on Moran's I, for the reference data (red line) and values for the same genes in the reconstructed data (blue line) (top). The gene-wise difference was computed as percentages and presented in a boxplot for these $N = 30$ top genes (bottom). **g** Investigation of additional new cells used for spatial reconstruction. Comparing the relative frequencies of cell types for original annotated cells (gray) and newly added cells (blue). **h** Spatial view of new cell's reconstructed spatial positions, related to the original reference cells (gray), for cell types selected based on distinct spatial patterns. All boxplots present median, upper and lower quartiles, with whiskers showing 1.5× interquartile range. DC, dendritic cell; mDC, myeloid dendritic cells; pDC, plasmacytoid dendritic cells. Data for all cell types are provided in the source data file. Source data are provided as a Source Data file.

In addition to spatial statistics performed on labeled data, such as cell type identities, we could also identify spatially variable genes based on gene expression levels and spatial distribution. By calculating the Moran's I measure of spatial autocorrelation for all genes, we could observe a high similarity between results from reference and reconstructed positions (Fig. 4f). From inspecting the top 30 most spatially variable genes in the reference data, the mean difference in Moran's I was about 4.1% with a Pearson correlation 0.986 for all genes (Fig. 4f, boxplot). As such, identification of the same spatially variable genes could be made with the reconstructed positions as one would with the Russell et al.-generated Slide-tags positions.

**Cell-bead network reconstruction allows for additional inclusion of biologically relevant cells**

Finally, we analyzed the re-introduced subset of 1989 cells for relevant biological information, i.e., the cells excluded from the original pipeline due to ambiguous spatial clustering but included in our cell-bead network-based reconstruction upon observing improved spatial accuracy. As these cells had been excluded from the original Slide-tags tonsil data analysis, they lacked annotations for cell type identities. Thus, after quality filtering, we performed a label transfer approach to annotate these new cells ($N = 1783$) by using the original cells as a reference (Methods). This allowed us to expand the dataset by more than 30% in terms of cell count. Overall, the cell type labels of the newly added cells roughly followed the same frequency distribution as the original cells, although an increase in the fraction of CD4+ T cells was observed (Fig. 4g). Moreover, by mapping the new cells spatially using their reconstructed coordinates, we could visually confirm that they coincide with their fellow cells of the same type (Fig. 4h).

In summary, the cell-bead network-based spatial reconstruction of the tonsil data not only faithfully captures the spatial biological architecture observed in the original Slide-tags study, but also extends its biological utility by enabling the inclusion of previously excluded cells. The reconstructed positions accurately preserve the cellular spatial niches and their biological context. Moreover, the cell-bead network approach allows us to include a larger number of cells and reduces data sparsity, thereby increasing the power to detect tissue organization and cell-cell relationships beyond what was possible in the reference-based dataset.

## Discussion

In this study, we revisited data from the Slide-tags method and uncovered a previously unexploited property: the barcode diffusion step produces a biophysical, bipartite cell-bead network that encodes spatial information. Although this network was not used for spatial reconstruction in the original method, it carries a latent spatial signal strong enough to recover cell positions. This phenomenon arises without any changes made to the original experimental protocol. On the contrary, for samples where full reconstruction is possible, it allows total omission of a major experimental step, the generation of a spatial reference map via optical decoding.

The fact that the pipeline can be used for accurate, reference-free spatial inference, places Slide-tags among a broader class of imaging-by-sequencing methods, techniques aiming to generate spatially informative molecular networks[10–18]. Within this family of techniques, the Slide-tags study is a unique case in which a ground-truth reference is available to directly verify the quality of network-based reconstruction, distinguishing this study from other imaging-by-sequencing demonstrations that relied on indirect methods of validation such as colocalization analysis. A parallel also exists with techniques like novoSpaRc[19,20], that infer spatial structure from expression similarity, with both our approach and theirs making use of latent spatial patterns in high-dimensional data. A recent follow-up by the developers of the Slide-tags method[21], demonstrates that by adjusting the chemistry of the bead array to contain both emitting and receiving bead barcodes, a diffusion-based network may be formed which encodes the spatial layout of the array. This approach parallels our observation that, without any experimental modification, such a bipartite network exists with cells as receivers. The two approaches, together, show that a network-based reframing of Slide-tags retains its original advantages, such as compatibility with single-nucleus sequencing, while unlocking the potential for optics-free spatial reconstruction from sequencing-data alone. Because spatial information is extracted from network topology rather than imaging, this decouples spatial resolution from array fabrication. As sequencing is required regardless, shifting spatial inference into the sequencing domain removes a major step that could enable larger scale substrates and higher-throughput applications of spatial transcriptomics.

Across the three datasets examined, we see that experimental factors can have a significant effect on the networks generated, and by extension their potential for and quality of spatial reconstruction. Although the total number of sequencing reads was similar, the resulting networks differed substantially in the number of edges or beads and ultimately in their signal-to-noise ratio (Supplementary Fig. 3, Supplementary Fig. 10). In the tonsil sample, the enrichment of spatially formed edges led to redundancy and a strong topological consensus that enabled accurate neighborhood characterization by our network structural discovery algorithm and subsequent spatial reconstruction. In contrast, the mouse embryonic brain and hippocampus samples contained a higher proportion of noise-derived or non-spatially correlated edges. This relationship can be seen in the cell-cell (unipartite) network projections (Fig. 2c), where the height of the edge frequency peak corresponding to close proximity connections relative to the background differs across samples. We speculate that such noisy edges could arise as a result of sample processing, e.g., unintended convection during the diffusion step, or errant probes during downstream nuclei handling.

Reconstruction quality improved further with the inclusion of originally omitted cells, i.e., cells that were excluded from the original Slide-tags analysis due to ambiguous clustering or weak spatial localization. This suggests a "wisdom of the crowd" mechanism of cooperative reinforcement whereby even cells that are not individually well-localized can, when included in aggregate, contribute usefully, if partially, to the spatial constraints giving rise to accurate reconstruction. In the case of the tonsil network, nearly half of the spatial structure was encoded in these previously unannotated cells. Their inclusion not only improved reconstruction metrics but also allowed for biologically meaningful label transfer and spatial mapping, ultimately expanding the analyzable dataset by over 30%.

A limitation of the cell-bead network approach is potentially that tissues with large empty regions, uneven cell distributions, or low network connectivity regions such as those present in the mouse hippocampus sample are less amenable to full-sample reconstruction. This reliance on underlying biophysical properties of the tissue is thus a potential shortcoming. For example, we observed a small difference in node degree as a function of cell type (Supplementary Fig. 11). Notably, while greater diffusion radius could hinder the assignment of spatial location in conventional Slide-tags (due to signal spread beyond the nearest cell-bead pair), it would likely benefit network-based reconstruction by increasing cell-bead overlap and overall spatial connectivity. This interestingly means that spread, which might be treated as a source of error in decoding-based methods, becomes a valuable feature in the network-based approach. We speculate that this could increase the robustness of the method by reducing the strict tolerances required for a successful experiment. Heterogeneous diffusion (Supplementary Fig. 12) could also affect network formation, although the redundant constraints created by overlapping diffusion clouds appear to compensate for these differences. Overall, it would

be expected that if adopted for large scale spatial characterization of tissue samples, the properties of individual tissues would need to be considered on a case-by-case basis (Supplementary Fig. 13).

Future work could further explore how experimental and biophysical parameters, including bead density, barcode design, and diffusion conditions, affect network formation and reconstructability. We anticipate that this strategy could extend to thicker or three-dimensional samples, where diffusion may help encode depth and spatial structure across tissue volumes. Overall, our findings show that spatial information encoded in a cell-bead molecular network can be decoded from sequencing data alone, providing a compelling framework for reference-free spatial transcriptomics.

## Methods

### Data retrieval
Reference positional data and gene expression data was retrieved through the Broad Institute Single-Cell Portal (SCP) via the accession numbers SCP2162 (mouse brain), SCP2170 (mouse embryonic brain), and SCP2169 (human tonsil) as deposited by Russell et al. Mouse sequencing data was acquired from the NCBI Sequencing Read Archive using the SRA Toolkit. As only the spatial sequences were of interest, only sequencing runs containing cell-bead paired reads were used, SRR26236611 and SRR26236607 for embryonic brain and mature hippocampus respectively. Spatial sequencing data for the human tonsil sample was provided by Russell et al. The data was processed using the code published by Russell et al., and the intermediate analysis file where cell-bead associations are clearly available was used for all further analysis.

### Initial network filtering
In the original Slide-tags procedure, beads with more than 256 UMIs were removed presumably due to their reflecting non-informative noise. Instead, the distribution of bead UMIs were first assessed, and an upper threshold was set for bead UMIs per sample corresponding to slightly above where the distribution significantly deviated from the power-law like appearance resulting in upper thresholds of 1500, 800, and 500 for tonsil, embryonic brain and hippocampus, respectively. All beads with only a single edge were also removed since they are not spatially informative in regard to relative positions (Supplementary Fig. 14).

### Spatial reconstruction of networks
Connected components obtained from the filtered unipartite cell-cell network were then spatially reconstructed using the Spatio-topological recovery by network discovery (STRND) pipeline[9]. The algorithm generated spatial representations of nodes based on their topological relationships within the network, resulting in spatial approximations for networks whose topology reflects physical spatial dependencies during formation.

First, random walks were conducted from each node in the graph to sample the visitation probabilities of nearby nodes. Let $V$ represent a graph of cell-bead nodes, and $P\left(v_i, v_j\right)$ denote the probability that node $v_j$ is visited during random walks initiated from the origin node $v_i$. These probabilities capture the local and global structure of the graph and are estimated from the visitation frequency gathered from random walks. These context walks were used as training data in a skip-gram architecture to produce high-dimensional representations $z_i$ for every node $v_i$[22]. The training was formulated as an optimization problem that aims to maximize the log-likelihood of observing the neighborhood $N(v_i)$ of each node given its vector representation $z_i$ (Eq. 1). The softmax function was used to map the pairwise vector representations into probability space and then frame it as an optimization problem where the goal is to minimize discrepancy between vector similarity and random walk visitation probabilities.

$$\mathcal{L} = \sum_{v_i \in V} \sum_{v_j \in N(v_i)} -log\left(\frac{exp\left(z_i^T z_j\right)}{\sum_{n_i \in V} exp\left(z_i^T z_n\right)}\right) \tag{1}$$

Thus, the structural information of the network was encoded through high-dimensional vector representations. UMAP was used to further reduce the dimensionality and map back to the original 2-dimensional space. This was done in such a way that if a pair of vectors $z_i, z_j$ are close in their high-dimensional representation (because of high random walk visitation probability), they will also be close in the mapped 2-dimensional space.

### Quantifying global reconstruction fidelity
The global reconstruction fidelity was quantified by computing the Correlation between Pairwise Distances (CPD), defined as the square of the Pearson correlation between the reference pairwise distances $d$ and the reconstructed distances $\hat{d}$ (Eq. 2). A high CPD indicates the preservation of distance patterns, meaning that the large-scale global configuration is accurately represented.

$$R_{d,\hat{d}}^2 = \left(\frac{\sum(d_i - \bar{d})(\hat{d}_i - \bar{\hat{d}})}{\sqrt{\sum(d_i - \bar{d})^2 \sum(\hat{d}_i - \bar{\hat{d}})^2}}\right)^2 \tag{2}$$

### Quantifying local reconstruction fidelity
For quantifying local accuracy, a KNN-based metric was used[9]. The original and reconstructed positions were used to construct two separate K-D trees. For each point the nearest neighbors were compared between the reference tree and the reconstructed tree. The overlap between them was assigned to each point as a score between 0 and 1 corresponding to the proportion of $K = 15$ nearest neighbors shared between trees. Here, $N_i^O(K) \cap N_i^R(K)$ and $N_i^O(K) \cup N_i^R(K)$ represent the intersection and union set of K-nearest neighbors of point i in the original and reconstructed datasets, respectively. The score assigned to the whole subgraph is the mean of the individual scores (Eq. 3).

$$\text{KNN score} = \frac{1}{n} \sum \frac{\left|N_i^O(K) \cap N_i^R(K)\right|}{\left|N_i^O(K) \cup N_i^R(K)\right|}. \tag{3}$$

The quality metric will vary with K, converging to 1 when K is equal to the number of nodes with a reference position in the network with a reference position (Supplementary Fig. 15).

### Quantifying shifts in position
To assess the reconstruction accuracy, the shift in position for the reconstruction once mapped to the original positions was characterized. This required aligning the reconstructed positions, $\hat{P}$, to the reference positions, $P$. This was accomplished using singular value decomposition (Eq. 4) and the point cloud norms, given by $||P|| = \sqrt{\Sigma\left|C_{i,j}\right|^2}$, to determine the transformation, $T$, for optimal alignment of reconstructed points on original points (Eq. 5).

$$U, \Sigma, V^T = svd(\hat{P}^T P) \tag{4}$$

$$T = UV^T \cdot \frac{||\hat{P}||}{||P||} \tag{5}$$

The distortion per point was then the Euclidean distance $d = \sqrt{(\hat{x} - x)^2 + (\hat{y} - y)^2}$, between the true position and corresponding aligned reconstructed point.

To correct for differences in point density, both the reference position and the reconstruction were normalized to a radius of 1 and centered at 0,0 followed by additional adjustment to match the positions of each convex hull prior to SVD calculation.

### Iterative bipartite reconstruction by DBSCAN-based edge filtering

To improve reconstruction quality, an iterative filtering strategy was applied to progressively remove long-range, noise-associated edges from the bipartite cell-bead network. This strategy was based on the observation that in high-quality spatial networks, the distribution of reconstructed edge lengths exhibits a dominant peak that corresponds to short-range spatial/proximity-driven connections, followed by a sharp decline. In contrast, noisier networks display a bimodal distribution, with one major narrow peak of short edges and a secondary peak associated with noise or non-spatially informative edges. Long range edges exerting an apparent compressive force and distortion source for the overall reconstruction were thus interpreted as a source of global noise that should be pruned from the reconstruction input data. To exploit this observation, DBSCAN-based clustering was used after a single-pass reconstruction to identify and retain short-range cell-bead associations prior to feeding the filtered input into STRND for subsequent re-reconstruction.

The procedure takes advantage of the fact that the cell-bead network reconstruction produces positions not only for cell-nodes but also bead-nodes. For each cell in the network, the reconstructed positions of all beads connected to that cell were extracted. DBSCAN clustering was then performed on these bead coordinates to identify spatially coherent subsets of beads associated with each individual cell. Beads not assigned to the dominant cluster for a given cell were interpreted as noise-associated connections. The corresponding edges between the cell and those beads were removed from the bipartite network, while the bead nodes themselves remained. The filtered network was then reprocessed using STRND to generate an updated spatial reconstruction. This selective edge pruning was aimed at eliminating the long-range, non-spatial associations progressively as the embedding becomes less distorted.

DBSCAN accepts two parameters, eps and min_samples. While eps control how far apart points can be to be assigned to a cluster, min_samples controls the minimum number of samples required to be counted as a cluster instead of noise. For this analysis, where scale is not invariant, eps was set to be 5% of the maximum reconstructed distance. On the other hand, min_samples was set to the value which maximizes the number of cells producing a single cluster of beads. For the case where a cell produced multiple clusters of beads, a simple heuristic of only preserving edges to beads in the largest cluster was followed. Furthermore, if a cell could not produce a cluster, that cell and its edges were removed from the network.

### Shape morphing using boundary-constrained adjustment

To improve the global accuracy of reconstructions, a custom shape-regularization script (Alphamorph) was developed. This method introduces prior information about sample shapes after an initial STRND reconstruction, using the reconstructed point cloud as input. The procedure first detects the reconstructed point cloud boundaries by using alpha shapes, and then applies a thin-plate spline (TPS) transformation to map the boundary to a predefined target shape (e.g., a circle in this case). The result is a point cloud that better represents the prior information on shape geometry (Supplementary Fig. 8).

First, the center of mass of the point cloud was computed, i.e., the centroid $(\hat{x}, \hat{y})$, by taking the average of all the points positions (Eq. 6):

$$\hat{x} = \frac{1}{N}\sum_i x_i, \hat{y} = \frac{1}{N}\sum_i y_i \qquad (6)$$

The radius $r$ was then defined as the maximum distance from this centroid to any point. This step provides a simple estimate of the overall size and center of the point cloud. After this, the boundaries of the point cloud are found via alpha shapes[23]. For a pair of points $p_1$ and $p_2$, the alpha shape retains an edge if there exists a circle with a radius $r' \le \frac{1}{\alpha}$ that contains no other points. The parameter $\alpha$ controls the level of detail, allowing the extraction of concave regions. The result is a set of boundary points that accurately represents the actual shape of the data.

These boundary points were later used as landmarks to create a correspondence between source and target shape. In particular, the direction between landmark and centroid is used to create target landmarks $(x', y')$ from the source landmarks (Eq. 7). Through this mapping, a one-to-one correspondence was established between the original boundary and the ideal circular boundary.

$$\varphi = arctan(y - \hat{y}, x - \hat{x}), (x', y') = (\hat{x} + r\cos\varphi, \hat{y} + r\sin\varphi). \qquad (7)$$

Once the source and target landmark pairs were obtained, a thin-plate spline (TPS) transformation[24] was computed. TPS is a smooth interpolation method that includes both a global linear transformation and a local non-linear warping component which minimizes the bending energy of the point cloud given target landmarks. The TPS transformation was then applied to every point in the original point cloud so that the overall envelope morphs to a circle while maintaining the relative point positions.

For accessibility, the method is publicly available in the form of a Python package, Alphamorph. https://github.com/DavidFernandez Bonet/alphamorph.

### Spatial analysis of human tonsil data

Downstream spatial analysis comparisons of the tonsil data were performed on the originally annotated cells ($n = 5631$) using the reference Slide-tags positions and the aligned reconstructed positions. The raw single nuclei gene expression data was first processed using Seurat[25] (v.4.4.0) in R (v.4.4.0), keeping only cells with reference and reconstructed positions. The data was normalized using the NormalizeData function with a scale factor of 10,000. The top 2000 variable genes were identified using FindVariableFeatures, using variance-stabilizing transformation (vst) correction, and thereafter, the gene expression was scaled using ScaleData. The Seurat object data was thereafter exported for analysis with Squidpy[26]. Using Squidpy (v.1.6.1) in Python (v.3.9.21), two AnnData spatial objects were set up for the reference and the reconstructed positions, which were used for downstream spatial analyses. The generation of spatial neighborhood graphs was first made using the squidpy.gr.spatial_neighbors() function. For the computation of neighborhood enrichment scores, the sq.gr.nhood_enrichment function was used, setting the cell type as the cluster_key argument. Co-occurrence scores for all cell types were computed using squidpy.gr.co_occurrence(), and later plotted by setting the cluster of interest to either 'B germinal center' or 'T CD4' cells. Ripley's statistics were computed for the cell types using the squidpy.gr.ripley() function, with mode set to "L" and max distance set to 2000. Computation of spatially variable genes was performed on the normalized expression for all available genes using Moran's I, by setting the mode to 'moran' in the squidpy.gr.spatial_autocorr function. All the final results presented in the main figure were plotted in R using ggplot2 (v.3.5.1).

## Cell type assignment of new cells in tonsil data

The human tonsil single nuclei data that lacked reference spatial positions, but could be assigned with reconstructed positions, were denoted as 'new' cells. The raw gene expression data of these new cells was processed using Seurat (v.4.4.0). Filtering was first performed to only include cells with a UMI count > 600, a unique gene count > 500, and a percentage of mitochondrial gene < 8.5% ($n = 1783$). These cutoff values were selected to match the original cell data. Thereafter, the new cell data was processed in the same way as the original cell data regarding normalization, finding variable features, and scaling. To assign cell type labels to the new cells, integration anchors for the filtered new cell data were first identified using FindTransferAnchors, with the original cell data as reference. Thereafter, cell type predictions were made using identified anchors and the original cell type labels. The cell type label with the highest prediction score was used to assign the predicted cell type for the new cells.

## Identifying and reconstructing mouse high connectivity regions

When creating the network unipartite cell-cell projects, a weight matrix $W$ is generated where each element represents the number of beads connected to both cells. A filtering process was implemented on the unipartite weight matrix $W$ to retain edges only if $W_{ij} \geq \theta$ where $\theta$ is the filtering threshold. As a result of thresholding, the network became sparser with the removal of connections, leading to the formation of multiple connected components, isolating subsets of cell nodes that remained strongly connected. A connected component in an unweighted graph is a subgraph $V \subseteq W$ in which any two nodes are connected to each other by paths, and which is connected to no additional nodes in the larger network.

A high threshold of $\theta = 7$ beads per edge resulted in subgraphs with a low mean degree of 2.2 for the 5 largest subgraphs. Spatial coherence[27] was used to inform an appropriate threshold, as the spatial coherence metric should be high and the network should separate into multiple subgraphs. When reconstructed, resulting positions have a high local accuracy, but global structure is largely lost. To rectify this, each subgraph $V \subseteq W$ goes through a process of extracting edges from lower filtering threshold $\theta_2$, where $\theta_2 < \theta_1$ but limited to only nodes within the subgraph established at $W_{ij} \geq \theta_1$. While this has the effect of also adding long edges that are not spatially precise, enough short-range, spatially accurate edges are able to counteract this. As the reconstruction algorithm is stochastic in nature, the relative abundance of these short, structurally beneficial edges overcomes the addition of long, destructive edges resulting in accurate reconstruction. When enriching a subgraph, the optimal $\theta_2$ is not guaranteed to be equal for all subgraphs, however, it was empirically observed that when including edges $\theta_2 = 1$, reconstructions were consistently of low quality, and therefore $\theta_2 \geq 2$ was used as a baseline. This observation could be attributed to edges composed of single beads being not only disproportionately numerous, but also appearing to be largely randomly formed. This approach, however, still had some requirements on spatial quality, and therefore was only possible when the subgraph connected cells were relatively evenly dispersed and was only possible if only cells with reference positions were included. (Supplementary Fig. 6).

## Bead diffusion modeling

To examine the statistical spatial relationship between beads and connected cell nodes, a relative analysis was conducted on the spatially reconstructed human tonsil dataset due to its high reconstruction quality. A dictionary was compiled of beads and their connections along with the post-reconstruction spatial position information for both cells and beads. Each bead location was placed at an origin and the relative location of each of the bead's connections were then plotted as a scatter around the origin. This was repeated for all beads, with information being added cumulatively to the scatter to obtain an overall relative spatial partner distribution for a typical bead in the specimen. The resulting profile resembled a superposition of a sharp and mostly-local Fickian diffusion profile and a broader distribution extending over the specimen (Supplementary Fig. 12).

The observed radial profile of bead-cell associations was modeled as the effective diffusion of DNA strands governed by Fick's second law in two dimensions, assuming an initial condition where all diffusing material is concentrated into a single point at time 0, to an expression for concentration as a function of radial distance from the origin bead and time (Eq. 8).

$$C(r, t) = \frac{M}{4\pi Dt} exp\left(-\frac{r^2}{4Dt}\right), C(r, t = 0) = M\delta(r), \quad (8)$$

Where $M$ is total material, and $D$ is the 2D diffusion coefficient and $(r)$ is the Dirac delta function.

It was hypothesized that the broader distribution represents non-spatially correlated connections, i.e., random connections to any location in the specimen. This would imply a distribution similar to the distribution of random line segments sampled from a disc, which has the formula (Eq. 9)

$$P(l) = \frac{4l}{\pi R^2}\left(\frac{l}{2R}\right) - \frac{2l^2}{\pi R^3}\sqrt{1 - \frac{l^2}{4R^2}}, \quad (9)$$

where $l$ is the length of a random line segment sampled from within a disc of radius $R$.

The diffusion model was combined with the line segment distribution and the result fitted to the radial profile of partner connections from an origin bead (Eq. 10), enabling extraction of a diffusion coefficient for the typical strand released from an origin bead.

$$h(r) = AP(r, R) + B exp\left(-\frac{r^2}{4Dt}\right), \quad (10)$$

where $A$ and $B$ are coefficients to fit.

Based on the methods described by Russell et al., photo-cleavage takes place for 30s or 3min, and incubation prior to cell extraction is 7.5min or 5min, depending on whether TAGS beads or SLAC beads were used, respectively. In both cases, the total time sums to 480s (8 min). Thus, substituting this time into the model and fitting to the radial relative position data, for the 104 nt Slide-Tags sequence release, diffusion, and capture in cell nuclei, the estimated diffusion coefficient is 26.22 $\mu m^2 s^{-1}$.

## Downsampling of the tonsil network

To examine how the properties of the bead-cell network of the tonsil sample affected reconstruction quality, the tonsil network was randomly downsampled in regard to the number of beads, number of edges, and number of total UMIs to the equivalent values of each mouse-derived network. The tonsil network was also downsampled in terms of cells, but as the mouse embryonic sample has a similar number, the cell-based downsampling was done to 5000 cells instead, in addition to the number of cells in the mouse hippocampus sample (Fig. 3b, Supplementary Fig. 2). Each property was used as the downsampling parameter five separate times for a total of 40 downsampled networks. Each network was then reconstructed five times to assess both sampling and reconstruction variance for a total of 25 network reconstructions per downsampling parameter and a total of 200 downsamplings (Supplementary Fig. 10).

### Additional datasets examined

We examined the Slide-tags sample from Hu et al., with bead-cell spatial reads accessed through SRR32297781. As no processed gene expression data was available publicly as in Russell et al. samples, gene expression reads were accessed through SRR32297780 and processed with Cell Ranger 6.1.2[28] to generate the cell barcode whitelist. The sample was then processed as the other samples, with exception of the quality metrics which were calculated using bead positions instead of cell positions, as only bead reference positions were available (Supplementary Fig. 16).

### Diffusion network simulation

A simulation aiming to recreate the properties observed in the sample network was created by initializing a surface of beads and cells in random positions across a 3 mm diameter circle. As the diffusion modeling was performed according to post-DBSCAN reconstruction 2D diffusion, the simulation adhered to this constraint in regard to edge formation. Each bead was allowed to form edges based on the probability derived from Eq. 8, and to recreate the trends visible in the empirical degree distributions each bead's edge formation was repeated several times governed by a random number sampled from a power-law distribution. Following this, a fraction of edges was replaced by a fully random edge according to a predetermined noise ratio (Supplementary Fig. 17).

### Reporting summary

Further information on research design is available in the Nature Portfolio Reporting Summary linked to this article.

## Data availability

Russell et al. produced Slide-tags datasets including reference positions and gene expression matrixes are available at the Broad Institute Single Cell Portal under the following accession numbers: SCP2162 (mouse brain), SCP2170 (mouse embryonic brain), SCP2169 (human tonsil). Russell et al. raw and processed mouse data are available at the Gene Expression Omnibus under accession number GSE244355. Data from Hu et al is available at the Broad Single Cell Portal under SCP2577 with sequencing data available from SRA under PRJNA1221542. Reconstructed position files generated in this study from Russell et al. data are provided in the Supplementary Source Data file. Larger files including full reconstructed positions and edgelists are available at https://doi.org/10.5281/zenodo.17013448 with exact filenames provided in the source data. Source data are provided with this paper.

## Code availability

The code used to develop the model, perform the analyses and generate results in this study is publicly available and has been deposited in Hidden_network_Slide_tags at https://github.com/molecular-programming-group/Hidden_network_Slide_tags, under MIT license. The specific version of the code associated with this publication is archived in Zenodo and is accessible via https://doi.org/10.5281/zenodo.17013448[29].

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

## Acknowledgements

We acknowledge funding from the Swedish Research Council (no. 2020-05368 to I.T.H.) and the European Research Council (no. 949624 to I.T.H.). The authors wish to thank Russell et al. and members of the Chen and Macosko groups for making their data publically available in addition to providing an additional dataset after inquiry and for the opportunity to study the results of their Slide-tags method.

## Author contributions

S.K.D., P.L.S., and I.T.H. conceived the study. S.K.D. and D.F.B. developed the core processing pipeline. L.F. performed the biological analysis. S.K.D., L.F., and I.T.H. developed and implemented visualization and analytical tools. All authors contributed interpretation, insights, and analyses of results. S.K.D. and I.T.H. wrote the manuscript with contributions from all authors.

## Funding

## Competing interests

L.F. is currently an employee at Pixelgen Technologies AB. I.T.H. is scientific advisor to and holds equity in a privately held startup that develops technologies related to sequencing-based inference. The remaining authors declare no competing interests.
