## [Transparent Peer Review file · Nature Communications]

Hidden network preserved in Slide-tags data allows reference-free spatial reconstruction

Corresponding Author: Dr Ian Hoffecker

Version 0:

Reviewer comments:

Reviewer #1

(Remarks to the Author)

The revised version addresses all my previous concerns and incorporates additional quality checks and benchmarking. These enhancements improve the clarity of the manuscript and better define the scenarios in which the method can be applied. I have no further comments.

(Remarks on code availability)

Reviewer #3

(Remarks to the Author)

I am satisfied with the revised manuscript and have no further comments.

(Remarks on code availability)

Reviewer #4

(Remarks to the Author)

My comments have been addressed. There is now a thoroughly documented jupyter notebook that guides through the initial analysis. It could maybe be linked from the README and overall it could be made more intuitive on "how to get started" but this more a matter of "user convenience".

I have also been asked to comment on Rev #2 comments: My impression is that these comments strive to push for a "perfect" method which is of course a valid goal, but often the perfect is the enemy of the good. The community will be served well by being able to use this method in addition to the reference-based analysis to "rescue" or improve - not using reference-free instead of reference-based. The limitations are in my opinion adequately described - it is clear that relatively homogenous spatial coverage is necessary and it likely won't work in low-quality data. As a compromise with Rev #2 one could maybe describe cases where it won't work more directly in the discussion, giving specific examples. But I agree with the authors that it is beyond the scope of this work to produce degraded samples/data only to show that it doesn't work...

(Remarks on code availability)

See comments to the authors.
